# The Ionotropic Receptors IR21a and IR25a mediate cool sensing in *Drosophila*

Lina Ni[1,2,3†], Mason Klein[4,5,6*†], Kathryn V Svec[1,2,3], Gonzalo Budelli[1,2,3], Elaine C Chang[1,2,3], Anggie J Ferrer[5], Richard Benton[7], Aravinthan DT Samuel[4,6*], Paul A Garrity[1,2,3*]

[1]National Center for Behavioral Genomics, Brandeis University, Waltham, United States; [2]Volen Center for Complex Systems, Brandeis University, Waltham, United States; [3]Department of Biology, Brandeis University, Waltham, United States; [4]Department of Physics, Harvard University, Cambridge, United States; [5]Department of Physics, University of Miami, Coral Gables, United States; [6]Center for Brain Science, Harvard University, Cambridge, United States; [7]Center for Integrative Genomics, Faculty of Biology and Medicine, University of Lausanne, Lausanne, Switzerland

*For correspondence: klein@miami.edu (MK); samuel@physics.harvard.edu (ADS); pgarrity@brandeis.edu (PAG)

†These authors contributed equally to this work

Competing interests: The authors declare that no competing interests exist.

**Abstract** Animals rely on highly sensitive thermoreceptors to seek out optimal temperatures, but the molecular mechanisms of thermosensing are not well understood. The Dorsal Organ Cool Cells (DOCCs) of the *Drosophila* larva are a set of exceptionally thermosensitive neurons critical for larval cool avoidance. Here, we show that DOCC cool-sensing is mediated by Ionotropic Receptors (IRs), a family of sensory receptors widely studied in invertebrate chemical sensing. We find that two IRs, IR21a and IR25a, are required to mediate DOCC responses to cooling and are required for cool avoidance behavior. Furthermore, we find that ectopic expression of IR21a can confer cool-responsiveness in an *Ir25a*-dependent manner, suggesting an instructive role for IR21a in thermosensing. Together, these data show that IR family receptors can function together to mediate thermosensation of exquisite sensitivity.

## Introduction

Temperature is an omnipresent physical variable with a dramatic impact on all aspects of biochemistry and physiology (*Sengupta and Garrity, 2013*). To cope with the unavoidable spatial and temporal variations in temperature they encounter, animals rely on thermosensory systems of exceptional sensitivity. These systems are used to avoid harmful thermal extremes and to seek out and maintain body temperatures optimal for performance, survival and reproduction (*Barbagallo and Garrity, 2015*; *Flouris, 2011*).

Among the most sensitive biological thermoreceptors described to date are the Dorsal Organ Cool Cells (DOCCs), a recently discovered trio of cool-responsive neurons found in each of the two dorsal organs at the anterior of the *Drosophila melanogaster* larva (*Klein et al., 2015*). The DOCCs robustly respond to decreases in temperature as small as a few millidegrees C per second (*Klein et al., 2015*), a thermosensitivity similar to that of the rattlesnake pit organ (*Goris, 2011*), a structure known for its extraordinary sensitivity. A combination of laser ablation, calcium imaging and cell-specific inhibition studies was used to establish the DOCCs as critical for mediating larval avoidance of temperatures below ~20°C, with the thermosensitivity of this avoidance behavior paralleling the thermosensitivity of DOCC physiology (*Klein et al., 2015*). While the DOCCs are

**eLife digest** Animals need to be able to sense temperatures for a number of reasons. For example, this ability allows animals to avoid conditions that are either too hot or too cold, and to maintain an optimal body temperature. Most animals detect temperature via nerve cells called thermoreceptors. These sensors are often extremely sensitive and some can even detect changes in temperature of just a few thousandths of a degree per second. However, it is not clear how thermoreceptors detect temperature with such sensitivity, and many of the key molecules involved in this ability are unknown.

In 2015, researchers discovered a class of highly sensitive nerve cells that allow fruit fly larvae to navigate away from unfavorably cool temperatures. Now, Ni, Klein et al. – who include some of the researchers involved in the 2015 work – have determined that these nerves use a combination of two receptors to detect cooling. Unexpectedly, these two receptors – Ionotropic Receptors called IR21a and IR25a – had previously been implicated in the detection of chemicals rather than temperature. IR25a was well-known to combine with other related receptors to detect an array of tastes and smells, while IR21a was thought to act in a similar way but had not been associated with detecting any specific chemicals. These findings demonstrate that the combination of IR21a and IR25a detects temperature instead.

Together, these findings reveal a new molecular mechanism that underlies an animal's ability to sense temperature. These findings also raise the possibility that other "orphan" Ionotropic Receptors, which have not been shown to detect any specific chemicals, might actually contribute to sensing temperature instead. Further work will explore this possibility and attempt to uncover precisely how IR21a and IR25a work to detect cool temperatures.

exceptionally responsive to temperature, the molecular mechanisms that underlie their thermosensitivity are unknown.

At the molecular level, several classes of transmembrane receptors have been shown to participate in thermosensation in animals. The most extensively studied are the highly thermosensitive members of the Transient Receptor Potential (TRP) family of cation channels (*Palkar et al., 2015*; *Vriens et al., 2014*). These TRPs function as temperature-activated ion channels and mediate many aspects of thermosensing from fruit flies to humans (*Barbagallo and Garrity, 2015*; *Palkar et al., 2015*; *Vriens et al., 2014*). In addition to TRPs, other classes of channels contribute to thermosensation in vertebrates, including the thermosensitive calcium-activated chloride channel Anoctamin 1 (*Cho et al., 2012*) and the two pore domain potassium channel TREK-1 (*Alloui et al., 2006*). Recent work in *Drosophila* has demonstrated that sensory receptors normally associated with other modalities, such as chemical sensing, can also make important contributions to thermotransduction. In particular, GR28B(D), a member of the invertebrate gustatory receptor (GR) family, was shown to function as a warmth receptor to mediate warmth avoidance in adult flies exposed to a steep thermal gradient (*Ni et al., 2013*). The photoreceptor Rhodopsin has also been reported to contribute to temperature responses, although its role in thermosensory neurons is unexamined (*Shen et al., 2011*).

Ionotropic Receptors (IRs) are a family of invertebrate receptors that have been widely studied in insect chemosensation, frequently serving as receptors for diverse acids and amines (*Benton et al., 2009*; *Silbering et al., 2011*). The IRs belong to the ionotropic glutamate receptor (iGluR) family, an evolutionarily conserved family of heterotetrameric cation channels that includes critical regulators of synaptic transmission like the NMDA and AMPA receptors (*Croset et al., 2010*). In contrast to iGluRs, IRs have been found only in Protostomia and are implicated in sensory transduction rather than synaptic transmission (*Rytz et al., 2013*). In insects, the IR family has undergone significant expansion and diversification, with the fruit fly *D. melanogaster* genome encoding 66 IRs (*Croset et al., 2010*). While the detailed structures of IR complexes are unknown, at least some IRs are thought to form heteromeric channels in which a broadly-expressed IR 'co-receptor' (such as IR25a, IR8a or IR76b) partners one or more selectively-expressed 'stimulus-specific' IRs (*Abuin et al., 2011*).

Among insect IRs, IR25a is the most highly conserved across species (*Croset et al., 2010*). In *Drosophila,* IR25a expression has been observed in multiple classes of chemosensory neurons with diverse chemical specificities, and IR25a has been shown to function as a 'co-receptor' that forms chemoreceptors of diverse specificities in combination with other, stimulus-specific IRs (*Abuin et al., 2011*; *Rytz et al., 2013*). IR21a is conserved in mosquitoes and other insects, but has not been associated with a specific chemoreceptor function (*Silbering et al., 2011*), raising the possibility that it may contribute to other sensory modalities.

Here, we show that the previously 'orphan' IR, *Ir21a,* acts together with the co-receptor IR25a to mediate thermotransduction. We show that these receptors are required for larval cool avoidance behavior as well as the physiological responsiveness of the DOCC thermosensory neurons to cooling. Furthermore, we find that ectopic expression of IR21a can confer cool responsiveness in an *Ir25a*-dependent manner, indicating that IR21a can influence thermotransduction in an instructive fashion.

## Results

### Dorsal organ cool cells express *Ir21a-Gal4*

To identify potential regulators of DOCC thermosensitivity, we sought sensory receptors specifically expressed in the dorsal organ housing these thermoreceptors (*Figure 1a*). Examining a range of potential sensory receptors in the larva, we found that regulatory sequences from the Ionotropic Receptor *Ir21a* drove robust gene expression (via the Gal4/UAS system [*Brand and Perrimon, 1993*]) in a subset of neurons in the dorsal organ ganglion, as well as in other locations (*Figure 1b,c*, *Figure 1—figure supplement 1*). Within each dorsal organ ganglion, *Ir21a-Gal4* drove gene expression in three neurons (*Figure 1b,c*). These neurons exhibited the characteristic morphology of the DOCCs, which have unusual sensory processes that form a characteristic 'dendritic bulb' inside the larva (*Klein et al., 2015*).

To confirm that the *Ir21a-Gal4*-positive neurons were indeed cool-responsive, their thermosensitivity was tested by cell-specific expression of the genetically encoded calcium indicator GCaMP6m under *Ir21a-Gal4* control. Consistent with previously characterized DOCC responses (*Klein et al., 2015*), when exposed to a sinusoidal temperature stimulus between ~14°C and ~20°C, GCaMP6m fluorescence in these neurons increased upon cooling and decreased upon warming (*Figure 1d,e* and *Figure 1—figure supplement 2*). The expression of *Ir21a-Gal4* was also compared with that of *R11F02-Gal4* (*Figure 1—figure supplement 1*), a promoter used in the initial characterization of the DOCCs (*Klein et al., 2015*). As expected, GCaMP6m expressed under the combined control of *Ir21a-Gal4* and *R11F02-Gal4* revealed their precise overlap in three cool-responsive neurons with DOCC morphology in the dorsal organ, further confirming the identification of the *Ir21a-Gal4*-expressing cells as the cool-responsive DOCCs (*Figure 1f,g*).

### *Ir21a* mediates larval thermotaxis

To assess the potential importance of *Ir21a* in larval thermosensation, we tested the ability of animals to thermotax when *Ir21a* function has been eliminated. Two *Ir21a* alleles were generated, *Ir21a^{123}* and *Ir21a^{Δ1}*. *Ir21a^{123}* deletes 23 nucleotides in the region encoding the first transmembrane domain of IR21a and creates a translational frameshift (*Figure 2a*). *Ir21a^{Δ1}* is an ~11 kb deletion removing all except the last 192 nucleotides of the *Ir21a* open reading frame, including all transmembrane and ion pore sequences (*Figure 2a*). As the deletion in *Ir21a^{Δ1}* could also disrupt the nearby *chitin deacetylase 5 (cda5)* gene (*Figure 2—figure supplement 1*), *Ir21a*-specific rescue experiments were performed to confirm all defects reflected the loss of *Ir21a* activity (see below).

The loss of *Ir21a* function strongly disrupted larval thermotaxis. When exposed to a thermal gradient of ~0.36°C/cm, ranging from ~13.5°C to ~21.5°C, *Ir21a^{Δ1}* null mutants as well as *Ir21a^{123}/ Ir21a^{Δ1}* heterozygotes were unable to navigate away from cooler temperatures and toward warmer temperatures (*Figure 2b,c*). These defects could be rescued by expression of a wild-type *Ir21a* transcript under *Ir21a-Gal4* control and by a wild-type *Ir21a* genomic transgene (*Figure 2c*). Taken together, these results are consistent with a critical role for *Ir21a* in larval thermotaxis.

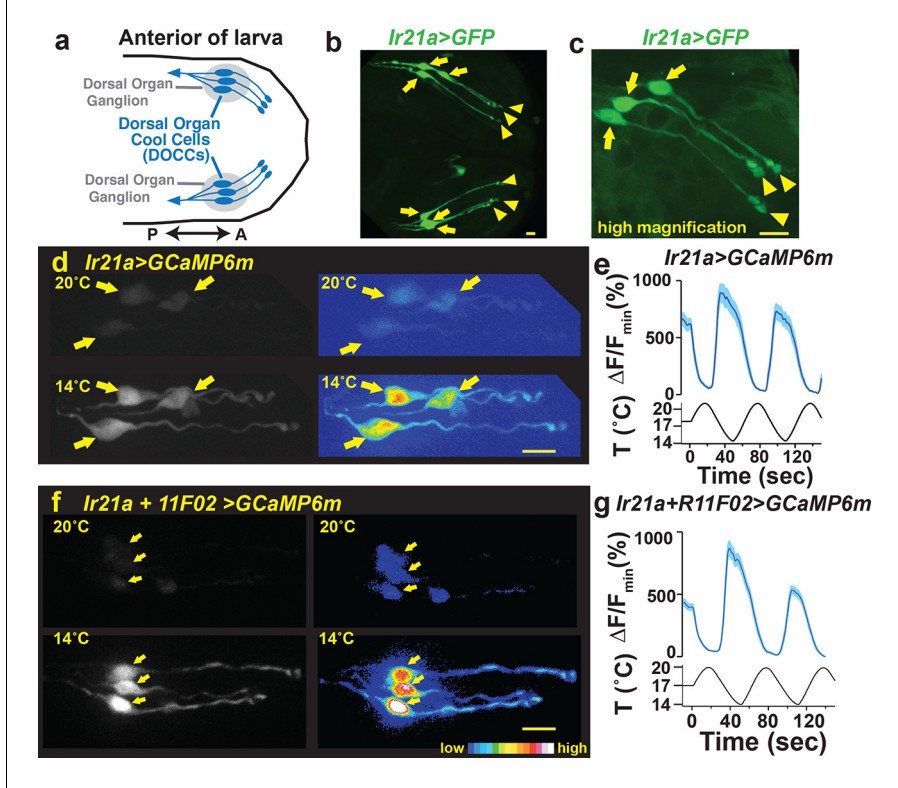

**Figure 1.** Dorsal Organ Cool Cells (DOCCs) express *Ir21a-Gal4*. (**a**) First/second instar larval anterior. Each Dorsal Organ Ganglion (grey) contains three DOCCs (blue). Anterior-Posterior axis denoted by double-headed arrow. (**b, c**) *Ir21a-Gal4;UAS-GFP* (*Ir21a>GFP*) labels larval DOCCs. Arrows denote cell bodies and arrowheads dendritic bulbs. (**d**) Temperature responses of *Ir21a-Gal4;UAS-GCaMP6m*-labeled DOCCs. Left panels, raw images; right panels, colors reflect fluorescence intensity. Arrows denote cell bodies. (**e**) Fluorescence quantified as percent change in fluorescence intensity compared to minimum intensity. n=22 cells (from 6 animals). (**f,g**) Temperature-responses of *Ir21a-Gal4;R11F02-Gal4;UAS-GCaMP6m*-labeled DOCCs. n=26 (7). Scale bars, 10 microns. Traces, average +/- SEM. *Figure 1—figure supplement 1* provides an example of the 3-D imaging stacks used for calcium imaging data acquisition.

The following figure supplements are available for figure 1:

**Figure supplement 1.** Larval-wide expression patterns of *Ir21a-Gal4* and *R11F02-Gal4*.

**Figure supplement 2.** Calcium-imaging data are obtained as a three-dimensional imaging stack.

## *Ir25a* mediates larval thermotaxis and is expressed in DOCCs

As IRs commonly act in conjunction with 'co-receptor' IRs, we examined the possibility that larval thermotaxis involved such additional IRs. Animals homozygous for loss-of-function mutations in two previously reported IR co-receptors, *Ir8a* and *Ir76b*, exhibited robust avoidance of cool temperatures, indicating that these receptors are not essential for this behavior (*Figure 2—figure supplement 1*). By contrast, $Ir25a^2$ null mutants failed to avoid cool temperatures, a defect that could be rescued by the introduction of a transgene containing a wild type copy of *Ir25a* (*Figure 2c*). Thus, *Ir25a* also participates in cool avoidance. To assess IR25a expression, larvae were stained with anti-sera for IR25a. Robust IR25a protein expression was detected in multiple cells in the dorsal organ ganglion, including the three *Ir21a-Gal4*-expressing DOCCs (*Figure 3a*). Within DOCCs, IR25a strongly labels the 'dendritic bulbs', consistent with a role in sensory transduction. Staining was

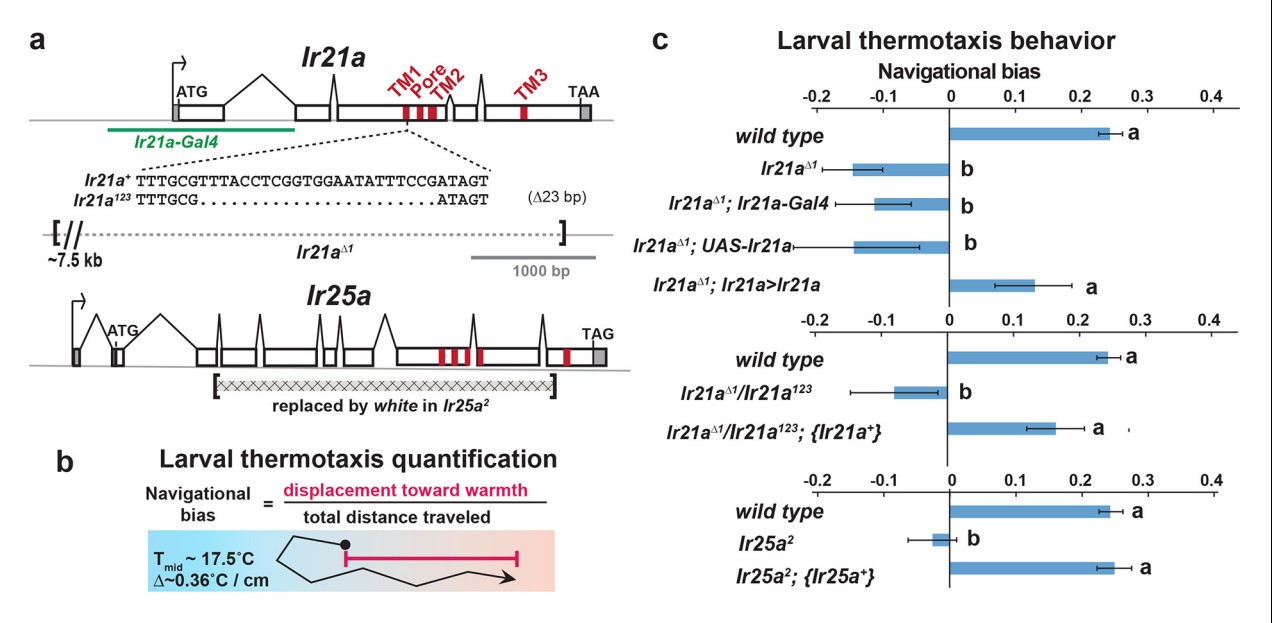

**Figure 2.** Larval cool avoidance requires *Ir21a* and *Ir25a*. (a) Sequence alterations in *Ir21a* and *Ir25a* alleles. *Ir21a* regulatory sequences present in *Ir21a-Gal4* are denoted in green and regions encoding transmembrane domains (TMs) and pore region in red. Additional details provided in *Figure 2—figure supplement 1*. (b) Thermotaxis is quantified as navigational bias. Cool avoidance behavior was assessed by tracking larval trajectories on a ~0.36°C/cm gradient extending from ~13.5°C to ~21.5°C, with a midpoint of ~17.5°C. (c) Cool avoidance requires *Ir21a* and *Ir25a*. *Ir21a>Ir21a* denotes a wild type *Ir21a* transcript expressed under *Ir21a-Gal4* control. {*Ir21a⁺*} and {*Ir25a⁺*} denote wild type genomic rescue transgenes. Letters denote statistically distinct categories (alpha=0.05; Tukey HSD). *wild type*, n=836 animals. $Ir21a^{\Delta 1}$, n=74. $Ir21a^{\Delta 1}$;*Ir21a-Gal4*, n=48. $Ir21a^{\Delta 1}$;*UAS-Ir21a*, n=10. $Ir21a^{\Delta 1}$;*Ir21a>Ir21a*, n= 88. $Ir21a^{\Delta 1}$/ $Ir21a^{123}$, n=71; $Ir21a^{\Delta 1}$/ $Ir21a^{123}$; {*Ir21a⁺*} n=70; $Ir25a^2$, n =100. $Ir25a^2$; {*Ir25a⁺*} n= 247. Additional mutant analyses provided in *Figure 2—figure supplement 1*.

The following figure supplement is available for figure 2:

**Figure supplement 1.** Structure of Ir21a locus and analysis of thermotaxis in Ir8a and Ir76b mutants.

absent in *Ir25a* null mutants demonstrating staining specificity (*Figure 3b*). Thus *Ir25a* is required for thermotaxis and is expressed in the neurons that drive this behavior.

## *Ir21a* and *Ir25a* are required for cool detection by DOCCs

To assess whether *Ir21a* and *Ir25a* contribute to cool detection by the DOCCs, DOCC cool-responsiveness was examined using the genetically encoded calcium sensor GCaMP6m. Consistent with a role for *Ir21a* in cool responses, DOCCs exhibited strongly reduced responses to cooling in $Ir21a^{\Delta 1}$ deletion mutants, and this defect was robustly rescued by expression of an *Ir21a* transcript in the DOCCs using *R11F02-Gal4* (*Figure 4a-e,h*). Similarly, DOCC thermosensory responses were greatly reduced in *Ir25a* mutants, a defect that was rescued by a wild type *Ir25a* transgene (*Figure 4f–h*). Together these data demonstrate a critical role for *Ir21a* and *Ir25a* in the detection of cooling by the DOCCs.

Prior work has suggested that three TRP channels, Brivido-1, Brivido-2 and Brivido-3, work together to mediate cool sensing in adult thermosensors (*Gallio et al., 2011*). Putative null mutations are available for two of these genes, *brv1* and *brv2*, and we used these alleles to test the potential role of Brivido function in DOCC cool sensing (*Gallio et al., 2011*). Although *brv1* mutant showed defects in thermotactic behavior, DOCC responses to cooling appeared unaffected in *brv1* mutants (*Figure 4—figure supplement 1*). *brv2* nulls exhibited no detectable thermotaxis defects (*Figure 4—figure supplement 1*). Thus, we detect no role for these receptors in cool sensing by the DOCCs.

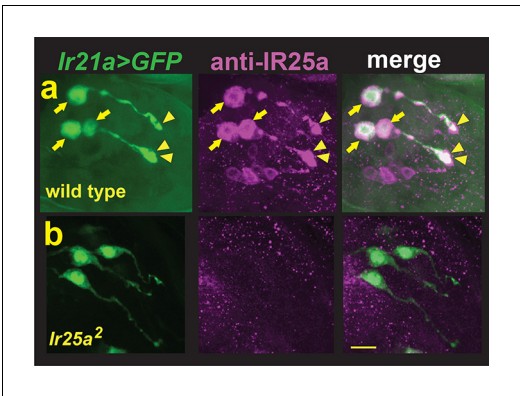

**Figure 3.** DOCCs express IR25a. (a) Left panel, *Ir21a>GFP*-labeled DOCCs. Middle panel, IR25a protein expression in dorsal organ. Right panel, *Ir21a>GFP*-labeled DOCCs express IR25a protein. Arrows denote DOCC cell bodies and arrowheads DOCC dendritic bulbs. (b) IR25a immunostaining is not detected in *Ir25a²* null mutants. Scale bar, 10 microns.

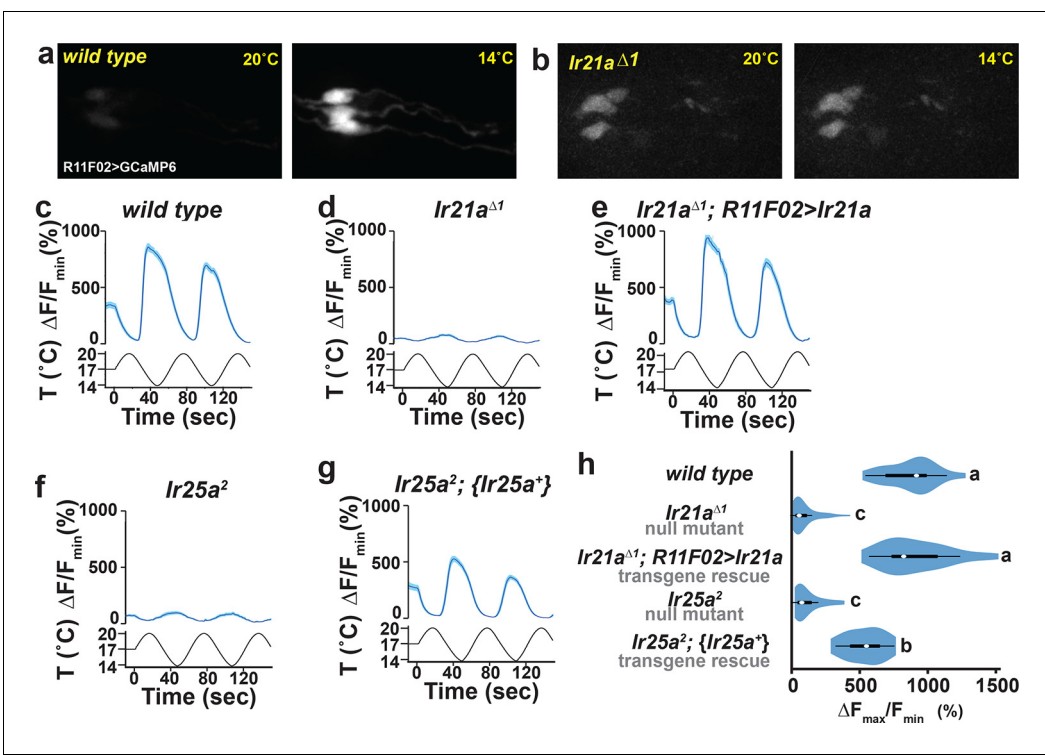

**Figure 4.** DOCC cool responses require *Ir21a* and *Ir25a*. DOCC responses monitored using *R11F02>GCaMP6m*. DOCCs exhibit robust cool-responsive increases in fluorescence (a,c), which are dramatically reduced in *Ir21a* (b,d) and *Ir25a* (f) mutants. (e) *Ir21a* transcript expression under *R11F02-Gal4* control rescues the *Ir21a* mutant defect. (g) Introduction of an *Ir25a* genomic rescue transgene rescues the *Ir25a* mutant defect. (h) Ratio of fluorescence at 14°C versus 20°C depicted using a violin plot. Letters denote statistically distinct categories, p<0.0001, Steel-Dwass test. Scale bars, 10 microns. Traces, average +/- SEM. *wild type*, n=33 cells (from 11 animals). *Ir21a^{Δ1}*, n= 58 (14). *Ir21a^{Δ1}; R11F02>Ir21a*, n=32 (9). *Ir25a²*, n=43 (16). *Ir25a²; {Ir25a^+}*, n=30 (10). Analyses of *brv1* and *brv2* mutants provided in *Figure 4—figure supplement 1*.
The following figure supplement is available for figure 4:

**Figure supplement 1.** Analysis of putative null mutants of *brv1* and *brv2*.

## Ectopic IR21a expression confers cool-sensitivity in an *Ir25a*-dependent manner

The requirement for *Ir21a* and *Ir25a* in DOCC-mediated cool sensing raised the question of whether ectopic expression of these receptors could confer cool-responsiveness upon a cell, as might be predicted for a cool receptor. Attempts to express IR21a and IR25a together or separately in heterologous cells, including S2 cells, *Xenopus* oocytes and HEK cells, failed to yield detectable responses to cooling or warming, as did attempts to confer thermosensitivity upon non-thermosensitive neurons by ectopically expressing them separately or together in *Drosophila*, broadly throughout the larval nervous system and in adult chemosensory neurons (G.B., L.N., M.K. and P.G, unpublished). However, ectopic expression of IR21a in one set of neurons in the adult, Hot Cell thermoreceptors in the arista that normally respond to warming rather than cooling, conferred cool-sensitivity.

The adult arista contains three warmth-activated thermosensory neurons, termed Hot Cells (or HC neurons) (*Gallio et al., 2011*). We found that forced expression of IR21a in the HC neurons could significantly alter their response to temperature. As previously reported (*Gallio et al., 2011*), wild-type HC neurons respond to warming with robust increases in intracellular calcium and to cooling with decreases in intracellular calcium, as reflected in temperature-dependent changes in GCaMP6m fluorescence (*Figure 5a,c*). In contrast, HC neurons in which IR21a is expressed under the control of a pan-neuronal promoter (*N-syb>Ir21a* animals) frequently exhibited elevations in calcium not only in response to warming, but also at the coolest temperatures (*Figure 5b,d,f*, *Figure 5—figure*

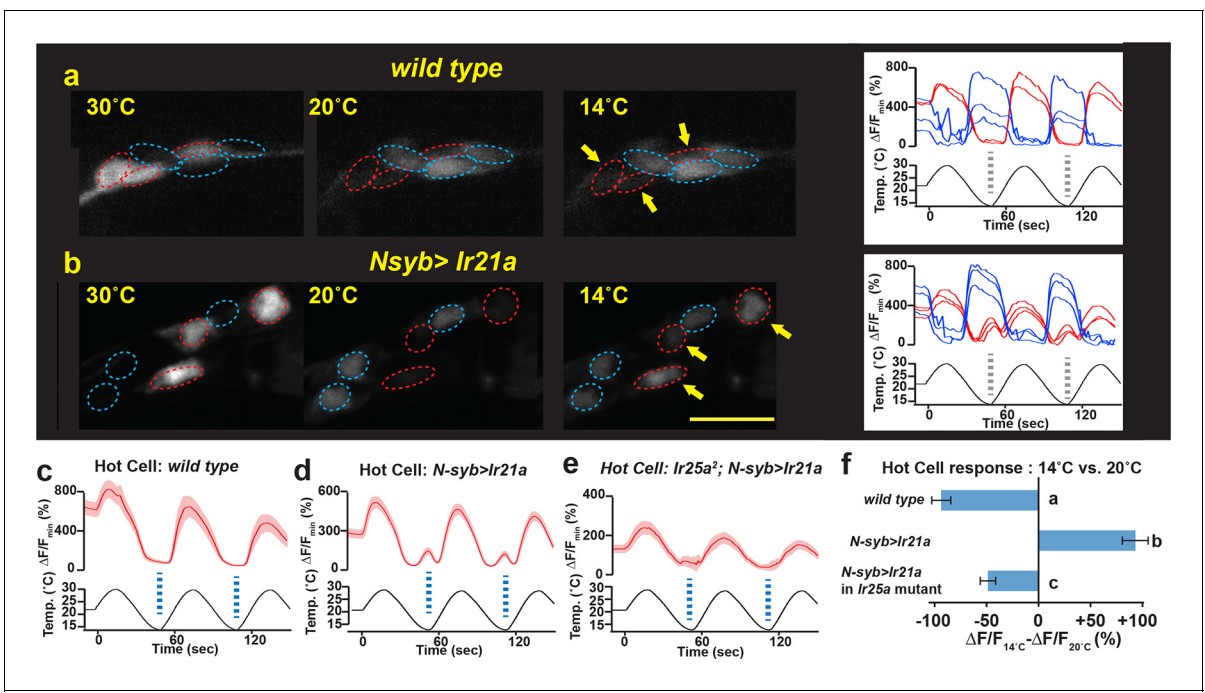

**Figure 5.** IR21a expression confers cool-sensitivity upon warmth-responsive Hot Cell neurons. (a,b) Temperature responses of *wild type* (a) or *N-syb>Ir21a*-expressing (b) thermoreceptors in the arista, monitored with *N-syb>GCaMP6m*. Cell bodies of warmth-responsive Hot Cells outlined in red and cool-responsive Cold Cells in blue. Arrows highlight Hot Cells at 14°C. Traces of Hot Cell and Cold Cell responses shown at right. Scale bar, 10 microns. (c-e) Fluorescence of Hot Cells in response to sinusoidal 14°C to 30°C temperature stimulus, quantified as percent ΔF/F$_{min}$. Dotted lines denote temperature minima. Traces, average +/- SEM. (f) Difference between ΔF/F$_{min}$ at 14°C vs 20°C (average +/- SEM). Responses of *N-syb>Ir21a* cells were statistically distinct from both *wild type* and *Ir25a²;N-syb>Ir21a* (p<0.01, Steel-Dwass test; letters denote statistically distinct groups). *wild type*, n= 16 cells (from 8 animals). *N-syb>Ir21a*, n= 16 (10). *Ir25a²; N-syb>Ir21a*, n= 20 (10). Analysis of endogenous IR25a expression in the Hot Cells and of the consequences of Hot Cell-specific misexpression of IR21a provided in *Figure 5—figure supplement 1*.

The following figure supplement is available for figure 5:

**Figure supplement 1.** Hot Cell neurons express IR25a protein, and IR21a confers cool-sensitivity upon the Hot Cell neurons.

*supplement 1a*). Thus, ectopic IR21a expression causes HC neurons, which are normally inhibited by cooling, to become responsive to both cooling and warming.

As *Ir21a*-dependent cool detection in the DOCCs relies upon *Ir25a*, we examined the requirement for *Ir25a* in IR21a-mediated cool activation of the HC neurons. Consistent with previously reported IR25a expression in the arista (*Benton et al., 2009*), we observed robust IR25a protein expression in the HC neurons (*Figure 5—figure supplement 1b,c*). Consistent with a role for *Ir25a* in *Ir21a*-mediated cool-responsiveness, ectopic IR21a expression failed to drive significant HC neuron cool responses in *Ir25a* mutants (*Figure 5e,f*). Thus, IR21a can confer cool-sensitivity upon an otherwise warmth-responsive neuron in an *Ir25a*-dependent fashion. Similar cool sensitivity was observed when IR21a was ectopically expressed under the control of an HC-specific promoter (*HC>Ir21a*, *Figure 5—figure supplement 1d,e*). Finally, ectopic expression of IR21a in *Gr28b* mutant HC neurons, which lack the Gr28b(D) warmth receptor, yields neurons that respond only to cooling (*Figure 6*). Together, these data demonstrate that ectopic IR21a expression can confer cool-sensitivity in an *Ir25a*-dependent fashion.

## Discussion

These data demonstrate that the ionotropic receptors IR21a and IR25a have critical roles in thermosensation in *Drosophila*, mediating cool detection by the larval dorsal organ cool cells (DOCCs) and the avoidance of cool temperatures. Combinations of IRs have been previously found to contribute to a wide range of chemosensory responses, including the detection of acids and amines (*Rytz et al., 2013*). These findings extend the range of sensory stimuli mediated by these receptor combinations to cool temperatures. Interestingly, IR21a- and IR25a-dependent cool sensation appears independent of Brivido 1 and Brivido 2, two TRP channels implicated in cool sensing in the adult (*Gallio et al., 2011*).

The precise nature of the molecular complexes that IRs form is not well understood. IR25a has been shown to act with other IRs in the formation of chemoreceptors, potentially as heteromers (*Rytz et al., 2013*). This precedent raises the appealing possibility that IR25a might form heteromeric thermoreceptors in combination with IR21a. However, our inability to readily reconstitute temperature-responsive receptor complexes in heterologous cells suggests that the mechanism by which these receptors contribute to cool responsiveness is likely to involve additional molecular cofactors. It is interesting to note that the range of cell types in which ectopic IR21a expression confers cool-sensitivity is so far restricted to neurons that already respond to temperature. This observation suggests the existence of additional co-factors or structures in these thermosensory cells that are critical for IR21a and IR25a to mediate responses to temperature. All studies to date implicate IRs as receptors for sensory stimuli (*Rytz et al., 2013*), and our misexpression studies are consistent with a similar role for Ir21a and IR25a in cool sensation. However, we cannot formally exclude the

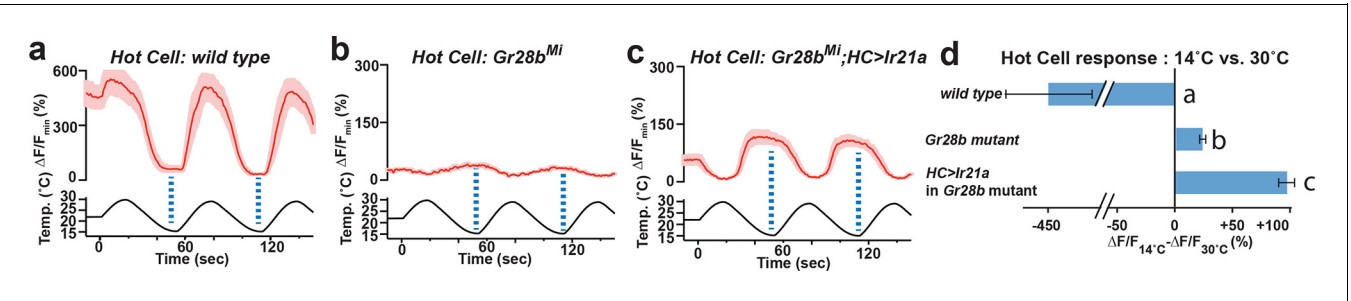

**Figure 6.** Hot Cell-specific expression of IR21a confers cool-sensitivity upon *Gr28b* mutant Hot Cell neurons. (a-c) Temperature responses of *wild type* (a), *Gr28b* mutant (b), and *HC>Ir21a*-expressing *Gr28b* mutant (c) thermoreceptors in the adult arista, monitored using *HC>GCaMP6m*. Dotted lines denote temperature minima. Traces, mean +/- SEM. *wild type*, n=11 cells (3 animals). *Gr28b*[Mi] n=9 (3). *HC>IR21a; Gr28b*[Mi] n=11 (3). (d) Cool responses ($\Delta F/F_{14°C}$ - $\Delta F/F_{30°C}$) of *HC>IR21a; Gr28b*[Mi] cells were distinct from both *wild type* and *Gr28b*[Mi] (p<0.01, Steel-Dwass test, letters denote statistically distinct groups).

possibility that they could have indirect, and possibly separate, functions in this process, for example, in regulating the expression or function of an unidentified cool receptor. Interestingly, IR25a was recently implicated in warmth-responsive resetting of the circadian clock, and suggested to confer warmth-sensitivity on its own, without the co-expression of other IRs (*Chen et al., 2015*). The ability of IR25a to serve as a warmth receptor on its own would be a surprise given both its broad expression and its established role as an IR co-receptor (*Abuin et al., 2011*). As IR25a misexpression only slightly enhanced the thermosensitivity of an already warmth-responsive neuron (*Chen et al., 2015*), this raises the alternative possibility that – analogous to cool-sensing – IR25a acts not on its own, but rather as a co-receptor with other IRs involved in warmth-sensing.

While the present study focuses on the role of IR21a and IR25a in larval thermosensation, it is interesting to note that the expression of both IR21a and IR25a has been detected in the thermoreceptors of the adult arista (*Benton et al., 2009*). Thus, related mechanisms could contribute to thermosensory responses not only in the DOCCs, but also in other cellular contexts and life stages. Moreover, the presence of orthologs of IR21a and IR25a across a range of insects (*Croset et al., 2010*) raises the possibility that these IRs, along other members of the IR family, constitute a family of deeply-conserved thermosensors.

## Materials and methods

### Fly strains

$Ir25a^2$ (*Benton et al., 2009>*), $BAC\{Ir25a^+\}$ (*Chen et al., 2015*), $Ir8a^1$ (*Abuin et al., 2011*), $Ir76b^1$ (*Zhang et al., 2013*), $Ir76b^2$ (*Zhang et al., 2013*), *R11F02-Gal4* (*Klein et al., 2015*), $brv1^{L653stop}$ (*Gallio et al., 2011*), $brv2^{w205stop}$ (*Gallio et al., 2011*), *HC-Gal4* (*Gallio et al., 2011*), $Gr28b^{Mi}$ (*Ni et al., 2013*), *UAS-GCaMP6m* (P{20XUAS-IVS-GCaMP6m}attp2 and P{20XUAS-IVS-GCaMP6m}attp2attP40 [*Chen et al., 2013*]), *UAS-GFP* (p{10X UAS-IVS-Syn21-GFP-p10}attP2 [*Pfeiffer et al., 2012*]), *nSyb-Gal4* (P{GMR57c10-Gal4}attP2, [*Pfeiffer et al., 2012*]), and y1 P(act5c-cas9, w+) M(3xP3-RFP.attP)ZH-2A w* (*Port et al., 2014*) were previously described.

In *Ir21a-Gal4*, sequences from -606 to +978 with respect to the *Ir21a* translational start site (chromosome 2L: 24,173 – 25757, reverse complement) lie upstream of Gal4 protein-coding sequences. *UAS-Ir21a* contains the *Ir21a* primary transcript including introns (chromosome 2L: 21823–25155, reverse complement) placed under UAS control. The {*Ir21a+*} genomic rescue construct contains sequences from -1002 to +4439 with respect to the *Ir21a* translational start site (chromosome 2L: 26153–20712).

$Ir21a^{\Delta 1}$ was generated by FLP-mediated recombination between two FRT-containing transposon insertions (PBac{PB}c02720 and PBac{PB}c04017) as described (*Parks et al., 2004*). $Ir21a^{123}$ was generated by transgene-based CRISPR-mediated genome engineering as described (*Port et al., 2014*), with an *Ir21a*-targeting gRNA (5'-CTGATTTGCGTTTACCTCGG) expressed under U6-3 promoter control (dU6-3:gRNA) in the presence of *act-cas9* (*Port et al., 2014*).

### Behaviour

Thermotaxis of early 2nd instar larvae was assessed over a 15 min period on a temperature gradient extending from 13.5 to 21.5°C over 22 cm (~0.36°C/cm) as described (*Klein et al., 2015*). As behavioral data appear normally distributed (as assessed by Shapiro-Wilk test), statistical comparisons were performed by Tukey HSD test, which corrects for multiple comparisons.

### Calcium imaging

Calcium imaging was performed as previously described for larvae (*Klein et al., 2015*). Pseudocolor images were created using the 16_colors lookup table in ImageJ 1.43r. Adult calcium imaging was performed as described for larvae (*Klein et al., 2015*), with modifications to the temperature stimulus and sample preparation approach. Adult temperature stimulus ranged from 14°C to 30°C. Intact adult antennae with aristae attached were dissected and placed in fly saline (110 mM NaCl, 5.4 mM KCl, 1.9 mM CaCl2, 20 mM NaHCO$_3$, 15 mM tris(hydroxymethyl)aminomethane (Tris), 13.9 mM glucose, 73.7 mM sucrose, and 23 mM fructose, pH 7.2, [*Brotz and Borst, 1996*]) on a large cover slip (24 mm x 50 mm) and then covered by a small cover slip (18 mm x 18 mm). The large cover slip was placed on top of a drop of glycerol on the temperature control stage. As quantified calcium imaging

data (*Figure 4h*, *Figure 5f*, *Figure 6d*) did not conform to a normal distribution as assessed by Shapiro-Wilk test (p<0.01), statistical comparisons were performed by Steel-Dwass test, a non-parametric test that corrects for multiple comparisons, using JMP11 (SAS).

## Immunohistochemistry

Immunostaining was performed as described (*Kang et al., 2012*) using rabbit anti-Ir25a (1:100; [*Benton et al., 2009*]), mouse anti-GFP (1:200; Roche), goat anti-rabbit Cy3 (1:100; Jackson ImmunoResearch), donkey anti-mouse FITC (1:100; Jackson ImmunoResearch).

## Acknowledgements

We thank Rachelle Gaudet and Linda Huang for comments on the manuscript, Guangwei Si for assistance with calcium imaging, Peter Bronk for advice on physiology, Adam Kaplan for creating *Ir21a-Gal4*, and the Bloomington Stock Center for fly strains. Supported by a grant from the National Institute of Neurological Disorders and Stroke (F32 NS077835) to MK, the National Institute of General Medical Science (F32 GM113318) to GB, European Research Council Starting Independent Researcher and Consolidator Grants (205202 and 615094) to RB, and the National Institute of General Medical Sciences (P01 GM103770) to ADTS and PAG.

## Additional information

### Funding

| Funder | Grant reference number | Author |
|---|---|---|
| National Institute of Neurological Disorders and Stroke | F32 NS077835 | Mason Klein |
| National Institute of General Medical Sciences | F32 GM113318 | Gonzalo Budelli |
| European Research Council | 205202 | Richard Benton |
| European Research Council | 615094 | Richard Benton |
| National Institute of General Medical Sciences | P01 GM103770 | Aravinthan DT Samuel Paul A Garrity |

The funders had no role in study design, data collection and interpretation, or the decision to submit the work for publication.

### Author contributions

LN, Performed molecular genetics, behavior, immunohistochemistry and calcium imaging, Conception and design, Acquisition of data, Analysis and interpretation of data, Drafting or revising the article, Contributed unpublished essential data or reagents; MK, Performed behavior and calcium imaging, Performed data analysis, Conception and design, Acquisition of data, Analysis and interpretation of data, Drafting or revising the article, Contributed unpublished essential data or reagents; KVS, ECC, Performed molecular genetics, Acquisition of data, Contributed unpublished essential data or reagents; GB, Performed physiology, Conception and design, Acquisition of data, Analysis and interpretation of data, Contributed unpublished essential data or reagents; AJF, Performed data analysis, Analysis and interpretation of data; RB, ADTS, Conception and design, Analysis and interpretation of data, Drafting or revising the article, Contributed unpublished essential data or reagents; PAG, Conception and design, Acquisition of data, Analysis and interpretation of data, Drafting or revising the article, Contributed unpublished essential data or reagents

### Author ORCIDs

Paul A Garrity, http://orcid.org/0000-0002-8274-6564

## Ethics

Animal experimentation: This study (specifically the harvest of *Xenopus laevis* oocytes) was performed in strict accordance with approved institutional animal care and use committee (IACUC) protocol (#14077) of Brandeis University.

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
