## [Decision Letter]

Thank you for submitting your work entitled "The Ionotropic Receptors IR21a and IR25a mediate cool sensing in *Drosophila*" for consideration by *eLife*. Your article has been favorably evaluated by K VijayRaghavan (Senior editor) and three reviewers, one of whom, (Ronald L. Calabrese) is a member of our Board of Reviewing Editors.

The reviewers have discussed the reviews with one another and the Reviewing Editor has drafted this decision to help you prepare a revised submission.

Summary:

This research report presents convincing genetic, calcium imaging, and behavioral data that specific Ionotropic Receptors (IRs) are required for cool sensing in Dorsal Organ Cool Cells (DOCCs) of the *Drosophila* larva and for cool avoidance (thermotaxis) in larvae. IR21a and IR25a appear to work together to confer cool sensing to the DOCCs. Moreover, ectopic expression of IR21a in thermosensitive neurons confers a component of cool sensitivity but requires IR25a expression. The demonstration that temperature sensing can be mediated by IRs has wide implication for the study of thermostatic behavior throughout the Protostomia, particularly insects.

The experiments are carefully performed and appropriately analyzed with relevant statistics. The Figures are easy to assimilate and the legends clear. Supplementary data answers for important controls. Materials and methods is adequate. Writing is very clear.

Essential revisions:

There are some concerns however that must be addressed before the manuscript can be published in *eLife*. While it is clear that IR21a and IR25a are necessary for cool sensing in DOCCs, it is not completely convincing that they themselves are cool receptors. As one expert reviewer points out "In contrast to earlier work showing that ectopic expression of *Drosophila* TRPA1 or GR28B(D) confers heat sensitivity to various cell types (e.g. Ni et al. Nature 2013 from the same group), this does not seem to be the case for IR21a and IR25a, except for a small response in the already thermosensitive Hot Cell neurons. Therefore, various other plausible scenarios can be envisaged where deletion of these proteins would abolish a cold response, without them being involved in the actual cold sensing. For instance, they could act downstream of the actual cold sensor (e.g. similar to a voltage-gated calcium channel at a sensory nerve ending), or their genetic elimination could merely influence the expression of the actual cold sensor. Similarly, it is suggested that these proteins act in a molecular complex, possibly as heteromultimers, but there is no evidence presented that these IRs interact with each other at the molecular level. Given that IR25a has already recently been implicated in thermosensing (Chen et al. 2015), I feel that, for this paper to represent a sufficient advance to merit publication in *eLife*, additional evidence supporting the hypothesis of IR21a/IR25a complexes as cold sensor should be provided. Minimally the authors should do an experiment where IR21a is expressed ectopically in thermosensitive neurons lacking a hot response (ectopic expression in a *Gr28* mutant) to show robust confirmation of cool sensing. Moreover, that authors should discuss their results in light of these caveats.

There are two other concerns from the expert reviewers that should be addressed.

1) What is the demonstration that the 3 sensory cells that respond to cool are responsible for the behavioral phenotypes? Complete expression patterns of *Ir21a-Gal4, Ir25a-Gal4*, and *R11F02* are necessary to evaluate this. If *Ir21a* or *R11F02* is only in the 3 cool cells, this is easily resolved. Otherwise, how do the authors determine that the sensory cells are responsible for the behavioral phenotype?

2) The BRV TRP channels have been implicated in sensing cool temperatures in adult *Drosophila* and it is unclear how these genes relate to the Irs reported here. From the 2015 PNAS paper, it appears that BRV1-*Gal4* is expressed in the cool-sensing neurons as well as additional sensory neurons in larvae. Separating out the role of these channel families in cool sensation would significantly advance the field and help resolve the different studies. Does a RNAi knock-down of BRV1,2, and 3 in *Ir21a* cells cause a behavioral or cellular phenotype? How do the authors resolve the loss of cool sensing in *brv1* mutant larvae with the lack of a cellular phenotype? Do other *brv1* neurons respond to cool? Do *brv1* neurons and IR21a neurons represent different cool cell populations or do they overlap in expression and function?

Minimally the authors should address this issue forthrightly in Discussion.

---

## [Author Response]

Essential revisions:

*There are some concerns however that must be addressed before the manuscript can be published in eLife. While it is clear that IR21a and IR25a are necessary for cool sensing in DOCCs, it is not completely convincing that they themselves are cool receptors. As one expert reviewer points out "In contrast to earlier work showing that ectopic expression of Drosophila TRPA1 or GR28B(D) confers heat sensitivity to various cell types (e.g. Ni et al. Nature 2013 from the same group), this does not seem to be the case for IR21a and IR25a, except for a small response in the already thermosensitive Hot Cell neurons. Therefore, various other plausible scenarios can be envisaged where deletion of these proteins would abolish a cold response, without them being involved in the actual cold sensing. For instance, they could act downstream of the actual cold sensor (e.g. similar to a voltage-gated calcium channel at a sensory nerve ending), or their genetic elimination could merely influence the expression of the actual cold sensor. Similarly, it is suggested that these proteins act in a molecular complex, possibly as heteromultimers, but there is no evidence presented that these IRs interact with each other at the molecular level. Given that IR25a has already recently been implicated in thermosensing (Chen et al. 2015), I feel that, for this paper to represent a sufficient advance to merit publication in eLife, additional evidence supporting the hypothesis of IR21a/IR25a complexes as cold sensor should be provided. Minimally the authors should do an experiment where IR21a is expressed ectopically in thermosensitive neurons lacking a hot response (ectopic expression in a Gr28 mutant) to show robust confirmation of cool sensing. Moreover, that authors should discuss their results in light of these caveats.*

As requested, we now demonstrate that Hot-Cell-specific *Ir21a* expression also confers cool sensitivity in a *Gr28b* null mutant. These new data are presented in a new Figure 6. The manuscript now states: “Finally, ectopic expression of Ir21a in *Gr28b* mutant HC neurons, which lack the Gr28b(D) warmth receptor, yields neurons that respond only to cooling (Figure 6).”

In terms of significance, it is important to note that while the Chen et al. paper implicated IR25a in thermosensing, our paper provides a very different view of IR25a’s role in the process. Chen et al. suggest IR25a can play an instructive role in thermosensing, acting as a warmth receptor capable of conferring warmth-sensitivity. Our data suggest that IR25a actually plays a permissive role, not specifically mediating hot, cold or chemical detection, but instead assisting other IRs responsible for mediating specific responses. Consistent with such a permissive role, our data show that IR25a also mediates cool sensing, suggesting that IR25a is not a warmth receptor, per se, but rather a molecule that can facilitate either warm or cold sensing depending on the cellular context. Further consistent with this notion, we show that IR25a does not act alone, but requires another IR to mediate cool sensation, and then show that misexpression of this other IR (IR21a) in the HC neurons can confer cool-sensitivity in an IR25a-dependent manner. Taken together, these data all support the notion that IR25a mediates thermosensing much like it mediates chemosensing, by assisting other IRs that have more cell-specific roles.

To put these data in context, most of what is known about IR25a is based on its role in chemical sensing. IR25a is expressed by hundreds of chemosensory neurons and does not appear to form functional chemoreceptors on its own. Instead, IR25a appears to play a permissive rather than an instructive role, serving as a co-receptor for odor-specific IRs that confer specificity for specific chemicals (Benton et al., 2009; Abuin et al., 2011; Silbering et al., 2011). Thus Chen et al.’s suggestion that Ir25a could confer warmth sensitivity when expressed on its own was unexpected. Chen et al. state that “IR25a misexpression confers temperature-dependent firing of heterologous neurons”, but the neurons they were recording from tonically fire in a warmth-responsive fashion even without IR25a misexpression (Chen et al., Figure 4I). IR25a misexpression increases the temperature co-efficient (Q10) of their spiking from 2 to 4 (Chen et al., Figure 4I). As warming speeds up most processes by similar margins (~90% of biological Q10s are between 1.3 and 5.1), this IR25a-dependent thermosensitivity is rather weak. As we note in our manuscript, the requirement for other IR subunits to mediate warmth detection could explain why misexpression of IR25a alone did not confer strong thermosensitivity.

We have adjusted the discussion of the Chen et al. paper slightly to attempt to emphasize this distinction. The manuscript now states: “Interestingly, IR25a was recently implicated in warmth-responsive resetting of the circadian clock, and suggested to confer warmth-sensitivity on its own, without co-expression of other IRs (Chen et al., 2015). The ability of IR25a to serve as a warmth receptor on its own would be a surprise given both its broad expression and its established role as an IR co-receptor (Abuin et al., 2011). As IR25a misexpression only slightly enhanced the thermosensitivity of an already warmth-responsive neuron (Chen et al., 2015), this raises the alternative possibility that – analogous to cool-sensing – IR25a acts not on its own, but rather as a co-receptor with other IRs involved in warmth-sensing. “

In contrast to earlier work showing that ectopic expression of Drosophila TRPA1 or GR28B(D) confers heat sensitivity to various cell types (e.g. Ni et al. Nature 2013 from the same group), this does not seem to be the case for IR21a and IR25a, except for a small response in the already thermosensitive Hot Cell neurons.

The size of the cool response may appear small because it is superimposed on a very large heat response (and cool inhibition). As shown in Figure 5, when cooled from 20°C to 14°C, wild-type HC neurons show an ~2-fold ∆F/F drop in GCaMP fluorescence, which Ir21a misexpression converts into an ~2 fold increase. While this is not as large as the HC neurons’ heat response, IR21a misexpression is turning a cold-inhibited cell into a cold-responsive cell, with an average ∆F/F of ~100%, similar to or larger than other published GCaMP6m responses. To help emphasize this, we now present the quantification of cool response data in a more traditional bar graph format (mean +/- SEM) in the main figure (Figure 5) and now present the individual data points in Figure 5—figure supplement 1. The impact of Ir21a misexpression is perhaps easier to see in the experiment requested by reviewers in Figure 6; in this figure, IR21a misexpression in a *Gr28b* mutant HC neuron yields a cool response in the absence of a heat response.

Therefore, various other plausible scenarios can be envisaged where deletion of these proteins would abolish a cold response, without them being involved in the actual cold sensing. For instance, they could act downstream of the actual cold sensor (e.g. similar to a voltage-gated calcium channel at a sensory nerve ending), or their genetic elimination could merely influence the expression of the actual cold sensor. Similarly, it is suggested that these proteins act in a molecular complex, possibly as heteromultimers, but there is no evidence presented that these IRs interact with each other at the molecular level.

We agree. While all work to date indicates that IRs act as sensory receptors and that IR25a functions only within heteromeric complexes, it would be exciting if this were not the case here. We have added a statement noting this: “All studies to date implicate IRs as receptors for sensory stimuli (Rytz et al., 2013), and our mis-expression studies are consistent with a similar role for Ir21a and IR25a in cool sensation. However, we cannot formally exclude the possibility that they could have indirect, and possibly separate, functions in this process, for example, in regulating the expression or function of an unidentified cool receptor.“

There are two other concerns from the expert reviewers that should be addressed.

*1) What is the demonstration that the 3 sensory cells that respond to cool are responsible for the behavioral phenotypes? Complete expression patterns of Ir21a-Gal4, Ir25a-Gal4, and R11F02 are necessary to evaluate this. If Ir21a or R11F02 is only in the 3 cool cells, this is easily resolved. Otherwise, how do the authors determine that the sensory cells are responsible for the behavioral phenotype?*

We should have been clearer in stating that this was previously addressed in Klein et al., 2015, PNAS. In the revised manuscript, we have altered the sentence in the Introduction that refers to this work. The original manuscript stated: “At the behavioral level, the DOCCs are critical for mediating larval avoidance of temperatures below ~20˚C, with the thermosensitivity of the avoidance behavior paralleling the thermosensitivity of DOCC physiology (Klein et al., 2015).” The revised manuscript now states: “A combination of laser ablation, calcium imaging and cell-specific inhibition studies were used to establish the DOCCs as critical for mediating larval avoidance of temperatures below ~20˚C, with the thermosensitivity of the avoidance behavior paralleling the thermosensitivity of DOCC physiology (Klein et al., 2015).”

To summarize, we showed that laser snipping the axons that connect the Dorsal Organs (DOs) to the brain totally eliminated cool avoidance (Klein et al., 2015; Figure 3). This showed both that the DO is essential for the behavior and, in addition, that sensory neurons outside the DO are insufficient to support even weak cool avoidance. Calcium imaging was then used to identify 3 cool sensing neurons within each DO (the DOCCs) and a Gal4 expressed in the DOCCs, R11F02-Gal4, was identified (Klein et al., 2015; Figure 3). We found that inhibiting R11F02-Gal4 neurons eliminated cool avoidance (Klein et al. Figure 3), and that optogenetically activating them elicited the same behaviors as cooling (Klein et el. Figure 5). In the current manuscript, we extend this prior work, identifying a second DOCC-expressed driver, Ir21a-Gal4 (see Figure 1, which demonstrates that Ir21a-Gal4 is expressed in the DOCCs but no other DO neurons), and find that expression of an Ir21a cDNA under Ir21-Gal4 control rescues cool avoidance. Taken together, the combined use of laser microsurgery and genetics establish the behavioral importance of the DOCCs.

While laser ablation studies indicate that sensory neurons outside the Dorsal Organ are insufficient to support cool avoidance, the requested full animal imaging of both R11F02-Gal4 and Ir21a-Gal4 expression is provided in Figure 1—figure supplement 1. (Ir25a-Gal4 was not used in the paper.). Outside the Dorsal Organ, both Gal4s are expressed by ~100 cells in the brain and ventral ganglion, neurons along the larval body wall and in the tail. R11F02-Gal4 is also expressed by sensory neurons in the Terminal Organ, which laser ablation shows is not required for cool avoidance behavior (Klein et al., 2015; Figure 3). Again, please note that the role of the Dorsal Organ (and hence the DOCCs) in the behavior does not rely on either R11F02-Gal4 or Ir21a-Gal4 being exclusively expressed in the Dorsal Organ, but was established using a combination of genetics and laser ablation.

2) The BRV TRP channels have been implicated in sensing cool temperatures in adult Drosophila and it is unclear how these genes relate to the Irs reported here. From the 2015 PNAS paper, it appears that BRV1-Gal4 is expressed in the cool-sensing neurons as well as additional sensory neurons in larvae.

NP4486-Gal4 (BRV1-Gal4) is a Gal4 enhancer trap inserted 2kb downstream of the Brv1 gene and 2.5kb upstream of the non-coding RNA gene CR32207 (http://flybase.org/reports/FBti0035982.html). Two recent RNA-seq studies suggest that NP4486-Gal4’s expression pattern appears to reflect CR32207 rather than Brv1 (Menuz et al. 2014 and Shiao et al. 2013). Specifically, while NP4486-Gal4 is widely expressed in the adult antenna (Gallio et al., 2011)), neither RNA-seq study detected any Brv1 transcripts in the antenna. Instead, both detected robust CR32207 expression. These data suggest that NP4486-Gal4 reports on CR32207. This is reasonable, as NP4486-Gal4 is inserted upstream of CR32207’s promoter, a common location for enhancer traps.

Separating out the role of these channel families in cool sensation would significantly advance the field and help resolve the different studies. Does a RNAi knock-down of BRV1,2, and 3 in Ir21a cells cause a behavioral or cellular phenotype? How do the authors resolve the loss of cool sensing in brv1 mutant larvae with the lack of a cellular phenotype? Do other brv1 neurons respond to cool? Do brv1 neurons and IR21a neurons represent different cool cell populations or do they overlap in expression and function?

Minimally the authors should address this issue forthrightly in Discussion.

We tested the roles of *brv1* and *brv2* in cool sensing by using previously published genetic null mutants rather than RNAi, as RNAi can have off-target effects and also tends to knock down rather than knock out the gene of interest. There are no Brv3 mutants. As shown in Figure 2—figure supplement 2, Brv1 null mutants have a cool avoidance defect, but their DOCC neurons still respond to cold normally. A mutation can cause a behavioral defect in many ways, but whatever Brv1’s role in cool avoidance, Brv1 is not required for DOCC cool sensing. Brv2 mutants show normal cool avoidance. Together our data indicate that IR-mediated cool sensing is independent of these receptors. The manuscript states: “Thus we detect no role for these receptors in cool sensing by the DOCCs.”

We have also added the following sentence to the Discussion:“Interestingly, IR21a- and IR25a-dependent cool sensation appears independent of Brivido 1 and Brivido 2, two TRP channels implicated in cool sensing in the adult (Gallio et al., 2011).”